# Tandem Electrospray Mass Spectrometry of Cyclic *N*-Substituted Oligo-β-(1→6)-D-glucosamines

**DOI:** 10.3390/ijms21218284

**Published:** 2020-11-05

**Authors:** Alexander O. Chizhov, Marina L. Gening, Yury E. Tsvetkov, Nikolay E. Nifantiev

**Affiliations:** N.D. Zelinsky Institute of Organic Chemistry, RAS, Leninskii Prosp., Moscow 119991, Russia; marina@mcl.ru (M.L.G.); tsvetkov@ioc.ac.ru (Y.E.T.); nen@ioc.ac.ru (N.E.N.)

**Keywords:** cyclooligosaccharides, glucosamine, glycoclusters, amides, artificial ion channels, glycoconjugates, fragmentation, electrospray, ionization, collisionally induced decay

## Abstract

High-resolution electrospray mass spectra (MS and MS/MS CID) of positive ions of a series of protonated, ammoniated, and metallated molecules of cyclic *N*-substituted oligo-β-(1→6)-D-glucosamines differing in cycle size and *N*-acyl substituents were registered and interpreted. It was shown that the main type of fragmentation is a cleavage of glycosidic bonds of a cycle, and in some cases fragmentation of amide side chains is possible. If labile fragments in substituents (e.g., carbohydrate chains) are present, a decay of the cycle and an elimination of labile fragments are of comparable possibility. It was found that in some cases rearrangements with loss of an internal carbohydrate residue (IRL), or an internal part of a side chain, are feasible.

## 1. Introduction

Cyclic oligosaccharides have attracted the attention of researchers for a long time. α-, β-, and γ-Cyclodextrins (i.e., cyclooligo-α-(1→4)-D-glucosides containing six, seven, and eight glucose chains, respectively) are the most well-known [1]. Previously, we synthesized cyclooligo-β-(1→6)-D-glucosamines having from two to seven monosaccharide residues and their *N*-acetyl derivatives, as the products of the terminated oligomerization of bifunctional monomers [2,3] and stepwise assembly of oligosaccharide chains [3,4,5,6]; their NMR spectra and conformational properties were studied [3,5,7]. In contrast to cyclodextrins, they are not prone to form inclusion complexes because of the absence of a hydrophobic cavity [7]. This feature allows one to use cyclooligo-β-(1→6)-D-glucosamines as matrices in the synthesis of versatile oligodentate molecular systems with preformed 3D spatial structure, due to the appropriate orientation of amino groups in cyclooligo-β-(1→6)-D-glucosamines in one direction from the planar section of a matrix [8,9]. Consequently, cyclooligo-β-(1→6)-D-glucosamines were used successfully in the design of lectin LecA blockers of the bacterium *Pseudomonas aeruginosa* [10] and in artificial ion channels [8,11].

Since the structural characterization of cyclooligo-β-(1→6)-D-glucosamines and related complex molecular systems by NMR is often impeded due to signal overlap [3,4,5,6,7], the development of a complementary approach based on mass spectrometry for investigation and/or support of the above structures is a realistic goal. There are several dozens of studies concerning the fragmentation of ions generated from cyclic oligosaccharides and their derivatives in a gas phase [12]. Previously, we studied the behavior of ionized cyclooligo-β-(1→6)-D-glucosamines and their *N*-acetyl derivatives in a gas phase under collision-induced dissociation (CID) [13]. Additionally, we carried out a preliminary study of isomeric effects in CID MS/MS of β-(1→6)-linked cyclic tetrasaccharides having Glc*p*_2_Glc*pN*_2_ composition (i.e., for the easiest example of heterogeneous cyclooligosaccharides); it was shown that there are substantial differences in MS/MS CID due to alternation of hexose and aminohexose units [14].

In the present paper, the peculiarities of electrospray ionization (ESI) high-resolution mass spectra (MS and MS/MS) of *N*-acyl derivatives of cyclooligo-β-(1→6)-D-glucosamines **1**–**24** (Figure 1) were studied using a hybrid instrument QqToF. This series includes cyclooligo-*N*-pentynoylglucosamines (**1**–**3**), cyclooligo-*N*-acylglucosamines bearing polyoxaalkyl chains (**4**–**13**, **19**–**22**), triazol linkers of different structures terminated with β-D-galactosyl residues (**14**–**16**, **24**), and polyoxaalkyl linkers terminated with β-D-galactosyl residues (**17**, **18**, **23**). The aim of this study was to search for basic regularities in the fragmentation of ions of the above compounds, such as cleavages of glycosidic bonds of cyclic structures, residues in cyclic structures and side chains, along with rearrangements.

## 2. Results and Discussion

ESI mass spectra of positively charged ions of cyclooligosaccharides **1**–**24** contain both singly and multiply charged ions, and protonated, ammoniated (NH_4_^+^), and metallated (Na^+^, K^+^) molecules having all possible combinations of associated ions (Appendix A). It was shown previously that there is a strong difference between protonated (ammoniated) molecules and their corresponding associates with sodium or potassium: the former undergo fragmentation at lower activation energies than the latter, and their fragmentation pathways often differ from each other [15]. Besides that, the higher the charge of an ion, the lower the activation energy needed for decay [16]. As usual, ammoniated and protonated molecules have similar CID MS/MS, since the primary process of decay of an ammoniated molecule can be explained as follows: [M+NH_4_]^+^ → [M+H]^+^ + NH_3_. Nevertheless, the formation of ammoniated fragments was observed; this phenomenon was observed for cyclic oligosaccharides in [17]. Note that isotopic clusters of [M+NH_4_]^+^ and [2M+2NH_4_]^2+^ overlap (peaks of monoisotopic ions coincide with one another due to the same *m*/*z* value; they are separable by ion mobility spectrometry, see [12] and references therein). Because non-covalent interaction between molecules can be regarded as weak, we have considered that the [2M+2NH_4_]^2+^ ion is transferred to two [M+NH_4_]^+^ under minimal impact, and thus the aforementioned overlap can be disregarded. In support of this approximation, we do not observe ions with higher *m*/*z* values than the initial [M+NH_4_]^+^ ion.

Tandem CID MS/MS of protonated and ammoniated molecules of cyclooligo-*N*-pentynoylglucosamines **1**–**3** are similar to those of cyclooligo-*N*-acetylglucosamines [13]. The main fragments are formed via the cleavage of glycosidic bonds, designated as [(GlcNHR_m_)_n_+H]^+^, where GlcN is a glucosamine residue, n is a number of residues, and R_m_ is an *N*-acyl substituent (in subscript arbitrary enumeration of substituents (m) is given)—series of small peaks are formed due to the additional elimination of water molecules. Note that for the MS/MS of cyclic oligosaccharides, fragment ions cannot be strictly classified using the generally accepted nomenclature [18]; see discussion in [12].

Negligible peaks in the low mass area may be assigned as decay products of hexosamine residue (*m*/*z* 176.0702 and *m*/*z* 164.0706), and elimination of an acyl fragment (*m*/*z* 144.0654). For MS/MS of cyclic tetramer **3**, see Figure 2.

Cleavages of glycosidic bonds were observed for [M+Na]^+^ ions of compounds **1**–**3**, however, fragmentations of pyranose rings were substantial; such behavior of metallated ions has been observed many times previously for acyclic oligosaccharides [15]. These fragments may be assigned as follows: *m*/*z* 324.1038, [GlcNHR_1_+C_2_H_4_O_2_+Na]^+^, *m*/*z* 408.270, [GlcNHR_1_+C_6_H_8_O_4_+Na]^+^, *m*/*z* 565.2008, [(GlcNHR_1_)_2_+C_2_H_4_O_2_+Na]^+^, *m*/*z* 625.2224 [GlcNHR_1_+C_4_H_8_O_4_+Na]^+^, and *m*/*z* 927.3466, [M–C_2_H_4_O_2_+Na]^+^) (Figure 3).

Similar results were obtained for compounds **4**–**7** containing two ether oxygen atoms in each of the *N*-acyl substituents: the main peaks in CID MS/MS corresponded to cleavages of glycosidic bonds, see for trimer **5** (Figure 4, [M+NH_4_]^+^ and Figure 5, [M+Na]^+^). In the MS/MS of the [M+NH_4_]^+^ ion, minor peaks due to the loss of one or two molecules of water were observed along with small peaks at *m*/*z* 318.1910 (C_15_H_28_NO_6_^+^) and *m*/*z* 707.4325 (C_34_H_63_N_2_O_13_^+^); the latter ones corresponded to the elimination of C_4_H_8_O, that is, the internal fragment of side chain from [(GlcNHR_2_)_n_+H]^+^ ions (n = 1, 2). The appearance of a peak at *m*/*z* 462.3062 (C_23_H_44_NO_8_^+^) can be also explained exclusively by transfer of the C_4_H_8_O fragment under CID. Previously, such rearrangement (transfer of oxaalkyl group) was not observed (nevertheless, there is a well-known transition of functional groups between carbohydrate residues; see the review on rearrangements of carbohydrate ions in mass spectra [19]). Fragmentation of the [M+Na]^+^ ion along with cleavage of glycosidic bonds accompanies the fission of *N*-acylated glucosamine residues (Figure 5).

In order to discover rearrangements, it seems interesting to investigate the CID of mixed cyclic tetramer **8**, in which *N*-acylated glucosamine residues (R_2_ = C_7_H_15_OC_4_H_8_OCH_2_CO) alternate with glucopyranose units. Indeed, attentive study of the CID MS/MS of [M+NH_4_]^+^ (Figure 6) revealed a very small (less than one rel. %) peak at *m*/*z* 779.4888 of the ion C_38_H_71_N_2_O_14_^+^ (calcd. *m*/*z* 779.4900), which corresponded to [(GlcNHR_2_)_2_+H]^+^ (Figure 6a, inset). Thus, the hypothesis of possible rearrangements of ions of cyclic oligosaccharides under CID being similar to internal residue loss (IRL) [19] was supported. The presence of ions *m*/*z* 480.2463 (C_21_H_38_NO_11_^+^) and *m*/*z* 1031.5394 (C_46_H_83_N_2_O_23_^+^) correlated with the loss of the internal ether fragment of a side chain C_4_H_8_O from [(GlcNHR_2_)Glc+H]^+^ and [M+H]^+^ ions, respectively. In the low-mass range (Figure 6b), one can find a small peak at *m*/*z* 318.1911 (C_15_H_28_NO_6_^+^) due to the loss of C_4_H_8_O from [GlcNHR_2_+H]^+^. The origination of other ions may be explained by simple cleavages of C–O-bonds of the side chain (*m*/*z* 171.1740, side chain, *m*/*z* 274.1293, elimination of the C_7_ fragment) and glucopyranose fragment (*m*/*z* 312.2164, C_17_H_30_NO_4_^+^, and *m*/*z* 324.2167, C_18_H_30_NO_4_^+^). CID MS/MS of the [M+Na]^+^ ion of compound **8** contained both products of the cleavage of glycosidic bonds, as well as pyranose units and side chains (Figure 7).

Compounds **9**–**12** are similar to **4**–**7**, but they possess shorter alkyl chains (methyl instead of heptyl). Their fragmentation under CID was also similar to that described above (primary cleavage of glycosidic bonds, fission of pyranose rings and side chains). CID MS/MS of the [M+H]^+^ ion of cyclic dimer **9** contained a peak of protonated unit [GlcNHR_3_+H]^+^
*m*/*z* 306.1545 and its fragmentation products (elimination of one or two water molecules, formaldehyde (the main peak), and so on, Figure 8). A peak at *m*/*z* 156.0658 arose due to rearrangement *m*/*z* 228→156, with the loss of the neutral fragment of the chain C_4_H_8_O. Fragmentation of an ammoniated molecule of **8** proceeded analogously. At elevated collision energy (60 eV), one can observe an ion *m*/*z* 87.0802 (C_5_H_11_O^+^), which corresponded to a fragment of side chain MeOC_4_H_8_^+^. A similar fragment was observed for ester MeO(CH_2_)_4_OCH_2_(CO)OMe, thus indirectly supporting the proposed mechanism (Appendix A).

Metallated ions of cyclooligomers **9**–**12** under CID gave ions which formed due to the cleavage of glycosidic bonds and the fragmentation of glucopyranose units. The doubly charged ion [M+H+Na]^2+^ of cyclic pentamer **12** was fragmented on both protonated and metallated singly and doubly charged ions (Figure 9).

The [M+NH_4_]^+^ ion of mixed cyclic tetramer **13** under CID gave mainly products of the cleavage of glycosidic bonds and the elimination of one of two water molecules from them (Figure 10). The peak of rearrangement product at *m*/*z* 611.3013 [(GlcNHR_3_)_2_+H]^+^ had negligible abundance (*ca*. 0.1%). The corresponding metallated ion [M+Na]^+^ gave fragments resulting from both the cleavage of glycosidic bonds and glucopyranose units (Figure 11). The ion [(GlcNHR_3_)_2_+Na]^+^ was not observed at all, and thus the absence of a rearrangement similar to IRL was shown.

Cyclic dimer **14** containing triazole heterocycle and galactosylated triethylene glycol linker in the side chain was synthesized by click reaction, presently widely used for the construction of biologically active conjugates [20]. The fragmentation of a doubly charged ion [M+2H]^2+^ proceeded by the elimination of two terminal galactosyl units (a peak at *m*/*z* 417.1982 had the highest intensity) and cleavage of glycosidic bonds of the cycle (Figure 12). A deglycosylated side chain (*m*/*z* 256.1282, C_11_H_18_N_3_O_4_^+^) gave a singly charged ion with the second highest abundance. Note that triazole heterocycle is resistant under CID conditions; elimination of N_2_ was not observed. Additionally, elimination of 1,2,3-triazole as whole (C_2_H_2_N_3_) did not occur. A scheme of the fragmentation of the [M+2H]^2+^ ion of cyclic dimer **14** under CID is proposed in Figure 13.

CID MS/MS of the [M+H+NH_4_]^2+^ ions of isomeric mixed, galactosylated cyclic tetramers **15** (**a**) and **16** (**b**) are given in Figure 14a,b, respectively. Acquisition was done under the same conditions. Because glucose and galactose are isomers (they are hexoses), it is impossible to determine which unit of C_6_H_10_O_5_ composition was eliminated by the use of CID MS/MS. Their spectra were very similar; small differences in the intensities of peaks were not conclusive. Main peaks arose due to glycosidic bond cleavages. Peaks of putative rearrangement ions were absent.

Cyclic trimer **17** and tetramer **18** have triethylene glycol galactosylated spacers in their side chains. Using **17** as an example, one can observe that fragmentation proceeded primarily due to cleavage of glycosidic bonds both for [M+2NH_4_]^2+^ (Figure 15) and [M+2Na]^2+^ (Figure 16). For the latter, charge separation with the elimination of Na^+^ and formation of the [M+Na]^+^ ion is possible along with covalent bond cleavages.

For cyclooligomers **19**–**22** having free hydroxyl groups at the ends of oligobutylene side chains (here the degree of oligomerization is equal to five), the fragmentation pattern changed drastically: along with glycosidic bond cleavage, fragmentation of side chains with the elimination of oligobutylene glycol fragments occurred along with the formation of singly and doubly charged ions (Figure 17 and Figure 18, respectively). A proposed scheme of the fragmentation of the [M+2NH_4_]^2+^ ion of cyclic tetramer **22** is presented in Figure 19. In these spectra, there were no ions which can be unequivocally regarded as the result of rearrangement.

An interesting result was obtained for ammoniated molecules [M+3NH_4_]^3+^, [M+2NH_4_]^2+^, etc. of cyclic trimer **23**, in which pentaethylene glycol side chains are galactosylated at the terminal hydroxyl groups (Figure 20). The expected fragments which were formed due to glycosidic bond cleavages (both terminal and cyclic) appeared protonated as well as ammoniated (cf. [17]). A possible explanation for the strong bonding of ammonium ions with fragments is the chelation of ammonium with oligoethylene glycol chains, which appeared resistant under low-energy CID. The [M+NH_4_+Na]^2+^ and [M+2Na]^2+^ ions underwent fragmentation analogously due to the cleavage of glycosidic bonds. Note that for the [M+NH_4_+Na]^2+^ ion, the formation of ammoniated fragments was not observed at 50 eV.

Mixed cyclic tetramer **24** possesses conformationally rigid side chains, containing triazole and *p*-aminophenyl fragments; the latter is *O*-galactosylated and *N*-acylated, that is, linked with an amide bond. As expected, the main fragmentation proceeded by the cleavage of glycosidic bonds (both terminal and cyclic, Figure 21). As mentioned above for **15** and **16**, the residues of isomeric hexoses in CID MS/MS are indistinguishable. Along with glycosidic bond cleavages, the fission of side chains with the elimination of the *p*-aminophenol residue C_6_H_7_NO was observed. We did not reveal rearrangement ions. A proposed scheme of the fragmentation of the [M+H+NH_4_]^2+^ ion of cyclic tetramer **24** under CID is presented in Figure 22.

We observed doubly charged ions of [M+Na+K]^2+^ composition for most of the compounds studied; their fragmentation resulted in the formation of the singly charged [M+Na]^+^ ion, whereas the peak of [M+K]^+^ had an intensity which was two orders of magnitude lower. We described this phenomenon previously in [21]. Apparently, the predominant formation of the [M+Na]^+^ ion may be explained by the lower ion radius of Na^+^ in comparison to K^+^, and hence a stronger ion–dipole interaction with a neutral analyte molecule.

## 3. Conclusions

In the course of the study of twenty-four cyclic *N*-substituted oligo-β-(1→6)-D-glucosamines, differing in cycle size and the nature of *N*-acyl substituents, general regularities of fragmentation of their protonated, ammoniated, and metallated molecules were revealed. It was shown that the main pathway of fragmentation is a cleavage of glycosidic bonds of a cycle, though in many cases the fission of side chains is possible (compounds **4**–**24**). When labile fragments are present in substituents (e.g., carbohydrate units, compounds **14**–**18**, **23**, and **24**), decay of the cycle and elimination of labile fragments from side chains are comparably possible. During fragmentation, rearrangements such as the elimination of an internal alkoxyl fragment from the side chain (compounds **4**–**13**) and IRL-like carbohydrate unit loss (compounds **8** and **13**) are also possible. The latter process is very minor. For metallated molecules [M+Met]^+^ (Met = Na, K), the cleavage of glycosidic bonds is accompanied with interlink cleavages of hexapyranose residues.

## 4. Materials and Methods

Compounds **1**–**24** (Figure 1) were synthesized previously [6,8,10,11]. Cyclooligosaccharides **1**–**3** and cyclic glycoconjugates **4**–**24** were dissolved in a mixture of acetonitrile and water, 50/50 vol. % before injection (LC-MS-grade acetonitrile was purchased from Merck, and HPLC-grade water was purchased from Sigma). High-resolution mass spectra (HRMS, R >30,000) were acquired on a maXis instrument (QqToF, Bruker Daltonics), using ESI in positive ion mode (capillary voltage was set at –4500 V) [22]. For compounds **1** and **2**, MS of negative ions were recorded (capillary voltage 2000 V). The registration range was from *m*/*z* 50 to *m*/*z* 3000 for MS, and from *m*/*z* 50 to *m*/*z* 1500 or *m*/*z* 3000 for MS/MS. Internal or external calibrations were applied using Electrospray Calibrant Solution (Fluka) or ESI Tuning Mix (Agilent). Syringe injection of solutions was used at 3 µL min^−1^. Nitrogen was used as a nebulizer and sheath gas (4 L min^−1^); interface temperature was set at 180 °C. Collision-induced dissociation mass spectra (CID) were acquired on a maXis instrument (nitrogen was used as a collision gas), activation energies were found in MS/MS experiment in real time and are given in figure legends. Standard tunes of orthogonal accelerator and other parts were set with a tune wide method for middle and high *m*/*z* values (higher than 300 Th), and a tune low method for low values, *m*/*z* from 50 to 300 Th. Formulas of primary and fragment ions were calculated according to their accurate masses using Bruker Compass 1.3 software.

## Figures and Tables

**Figure 1 ijms-21-08284-f001:**
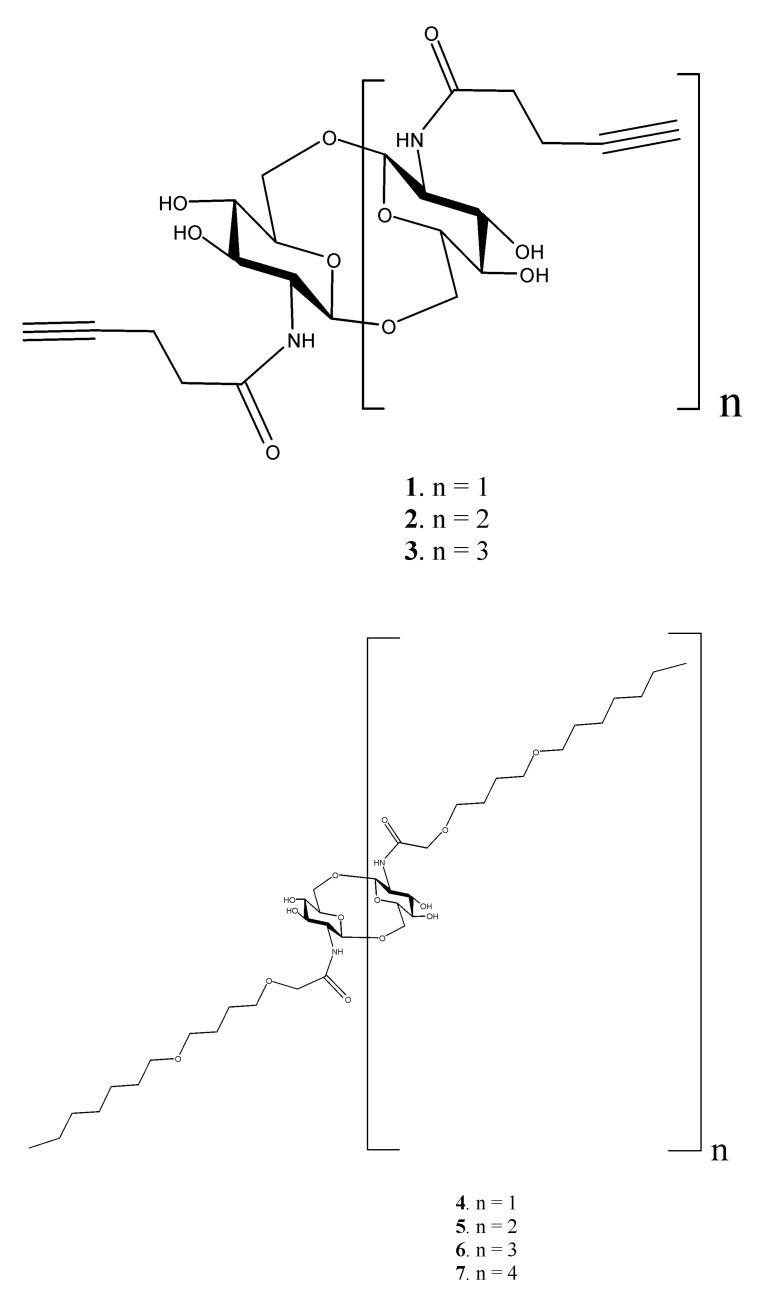
Structures of *N*-acylated derivatives of cyclooligo-β-(1→6)-D-glucosamines **1**–**24** studied in this work (continued on pages 3–7).

**Figure 2 ijms-21-08284-f002:**
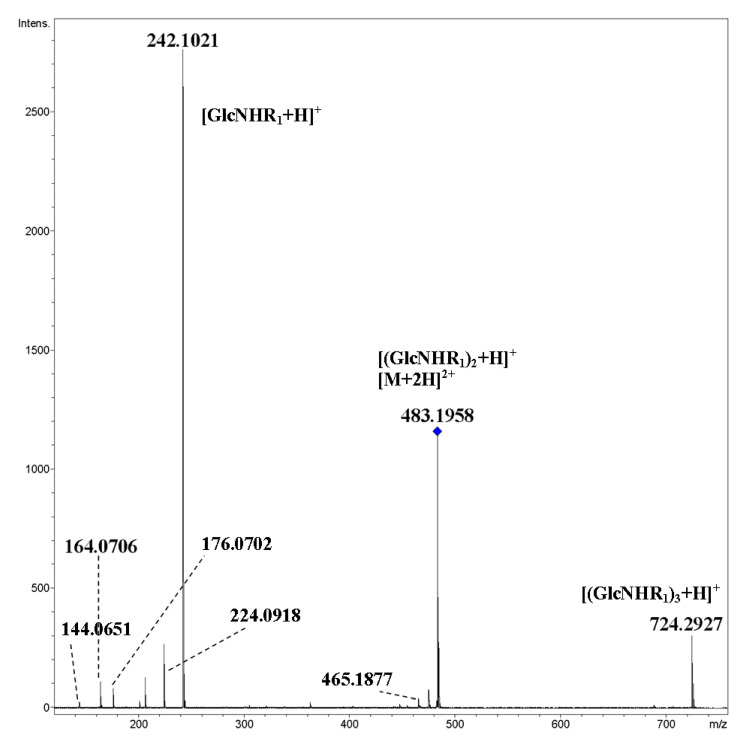
CID MS/MS of the [M+2H]^2+^ ion of cyclic tetramer **3** (M 964.38 Da, R_1_ = HCCCH_2_CH_2_CO), *m*/*z* 483, E_a_ 10 eV, tune low acquisition method.

**Figure 3 ijms-21-08284-f003:**
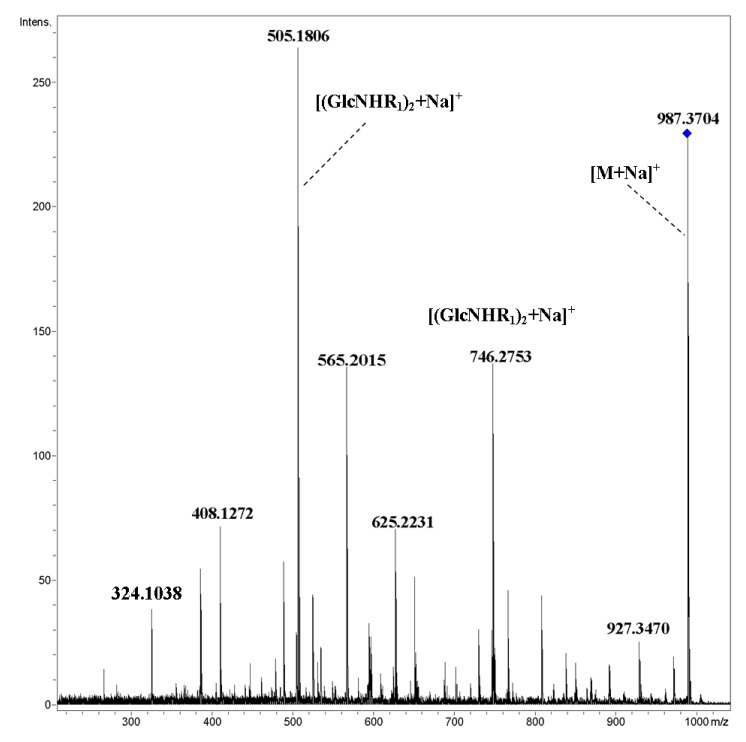
CID MS/MS of the [M+Na]^+^ ion of cyclic tetramer **3** (M 964.38 Da, R_1_ = HCCCH_2_CH_2_CO), *m*/*z* 987, E_a_ 80 eV, tune wide acquisition method.

**Figure 4 ijms-21-08284-f004:**
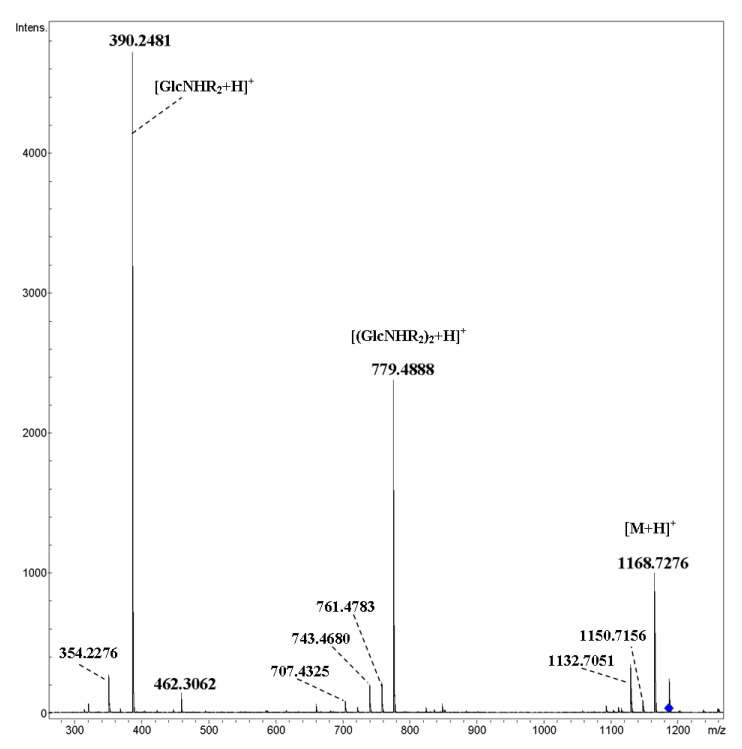
CID MS/MS of the [M+NH_4_]^+^ ion of cyclic trimer **5** (M 1167.72 Da) (R_2_ = C_7_H_15_OC_4_H_8_OCH_2_CO), *m*/*z* 1186, E_a_ 35 eV, tune wide acquisition method.

**Figure 5 ijms-21-08284-f005:**
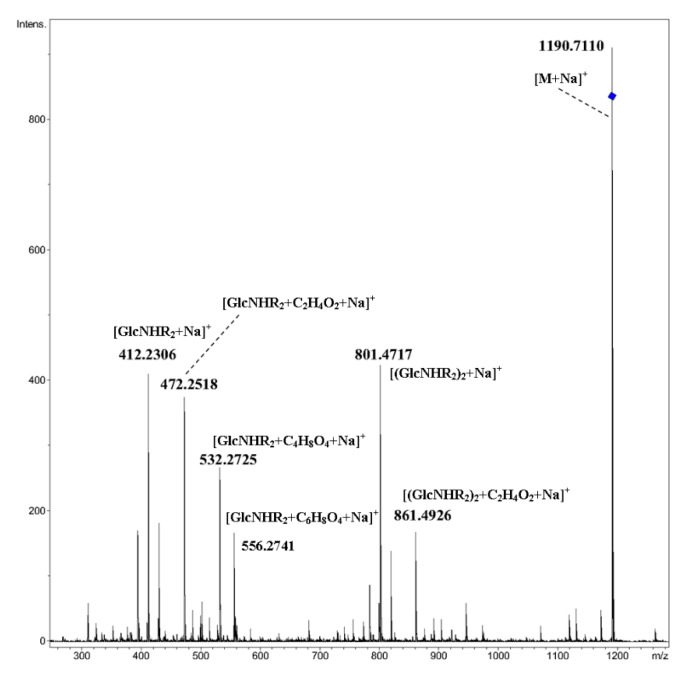
CID MS/MS of the [M+Na]^+^ ion of cyclic trimer **5** (M 1167.72 Da, R_2_ = C_7_H_15_OC_4_H_8_OCH_2_CO), *m*/*z* 1186, E_a_ 100 eV, tune wide acquisition method.

**Figure 6 ijms-21-08284-f006:**
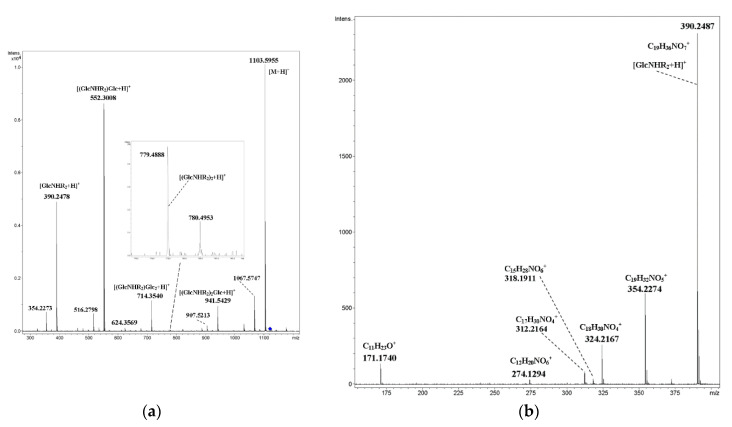
CID MS/MS of the [M+NH_4_]^+^ of mixed, cyclic tetramer **8** (M 1102.59, R_2_ = C_7_H_15_OC_4_H_8_OCH_2_CO), *m*/*z* 1121. (**a**) E_a_ 35 eV, tune wide acquisition method. (**b**) E_a_ 40 eV, tune low acquisition method (low mass range, *m*/*z* 150–400).

**Figure 7 ijms-21-08284-f007:**
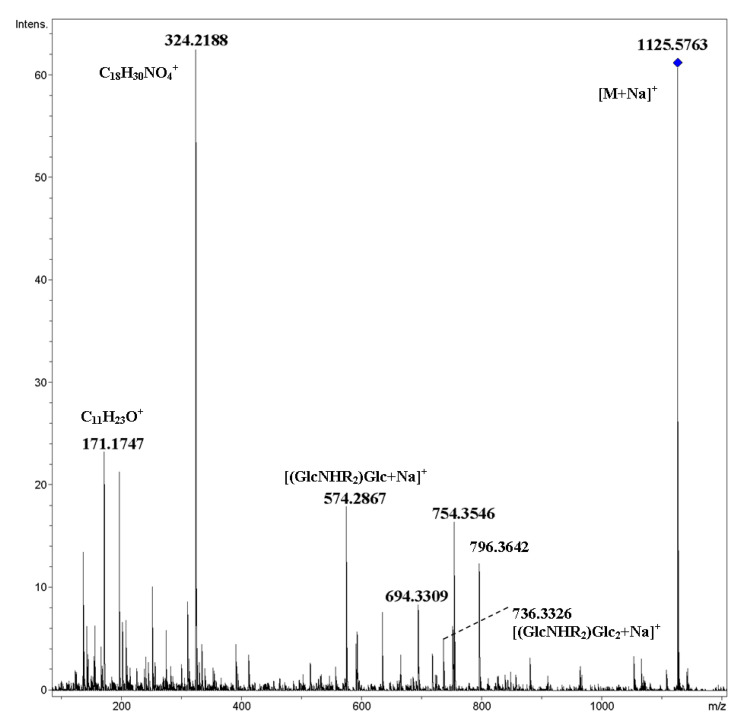
CID MS/MS of the [M+Na]^+^ ion of mixed, cyclic tetramer **8** (M 1102.59, R_2_ = C_7_H_15_OC_4_H_8_OCH_2_CO), *m*/*z* 1126, E_a_ 90 eV, tune low acquisition method.

**Figure 8 ijms-21-08284-f008:**
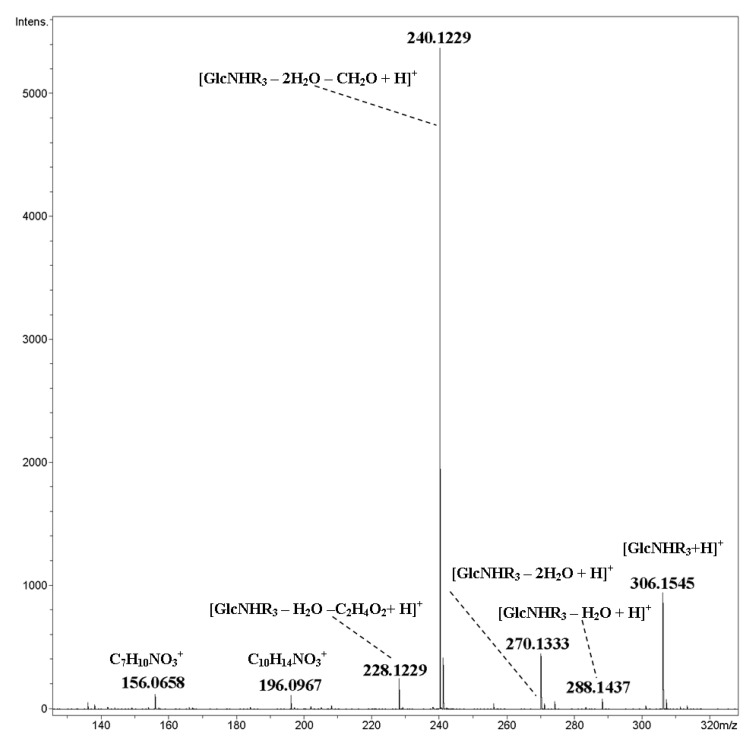
CID MS/MS of the [M+H]^+^ of cyclic dimer **9** (M 610.29 Da, R_3_ = CH_3_OC_4_H_8_OCH_2_CO), *m*/*z* 611, a range of *m*/*z* 130–325 is shown, E_a_ 35 eV, tune low acquisition method.

**Figure 9 ijms-21-08284-f009:**
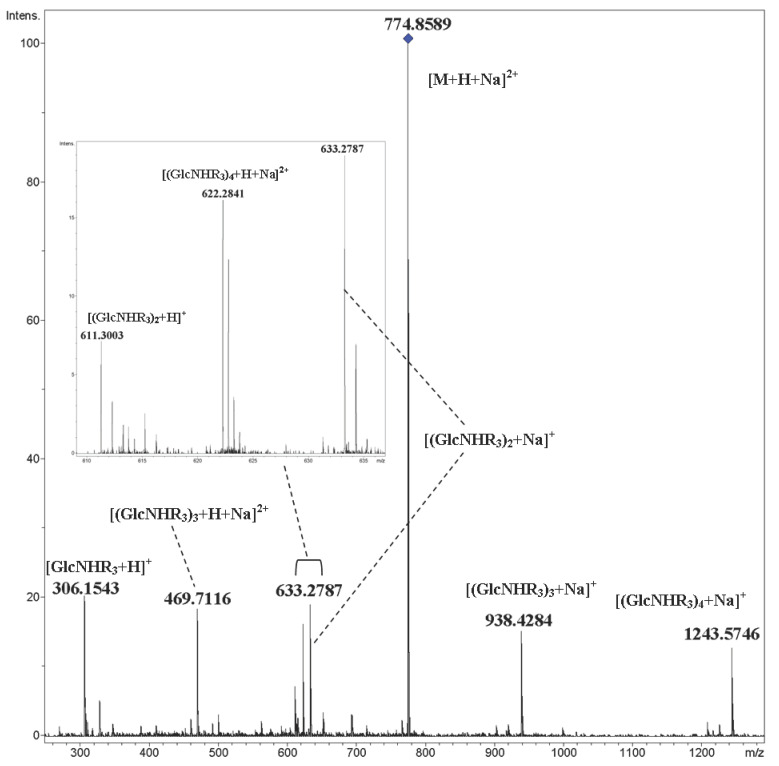
CID MS/MS of the [M+H+Na]^2+^ ion of cyclic pentamer **12** (M 1525.74, R_3_ = CH_3_OC_4_H_8_OCH_2_CO), *m*/*z* 775, E_a_ 30 eV, tune wide acquisition method. Inset: extension of *m*/*z* 610–638 range.

**Figure 10 ijms-21-08284-f010:**
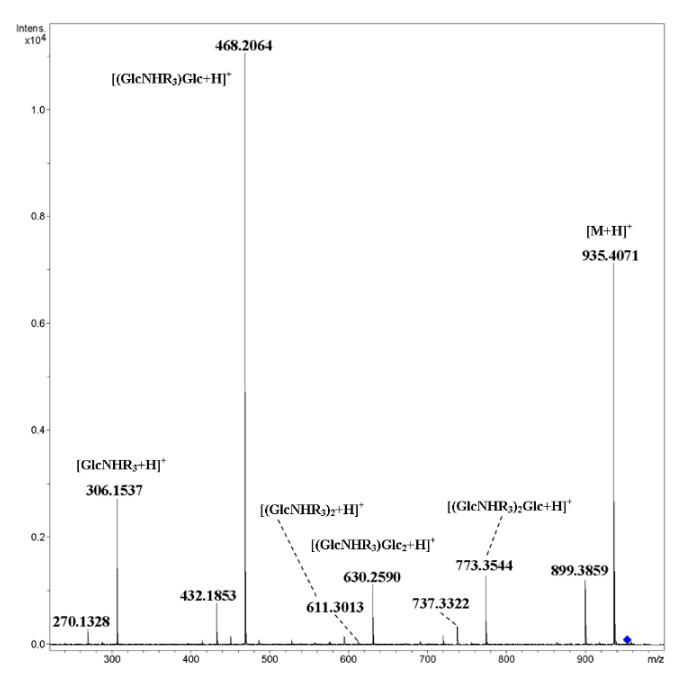
CID MS/MS of the [M+NH_4_]^+^ ion of mixed, cyclic tetramer **13** (M 934.40 Da, R_3_ = CH_3_OC_4_H_8_OCH_2_CO), *m*/*z* 952, E_a_ 35 eV, tune wide acquisition method.

**Figure 11 ijms-21-08284-f011:**
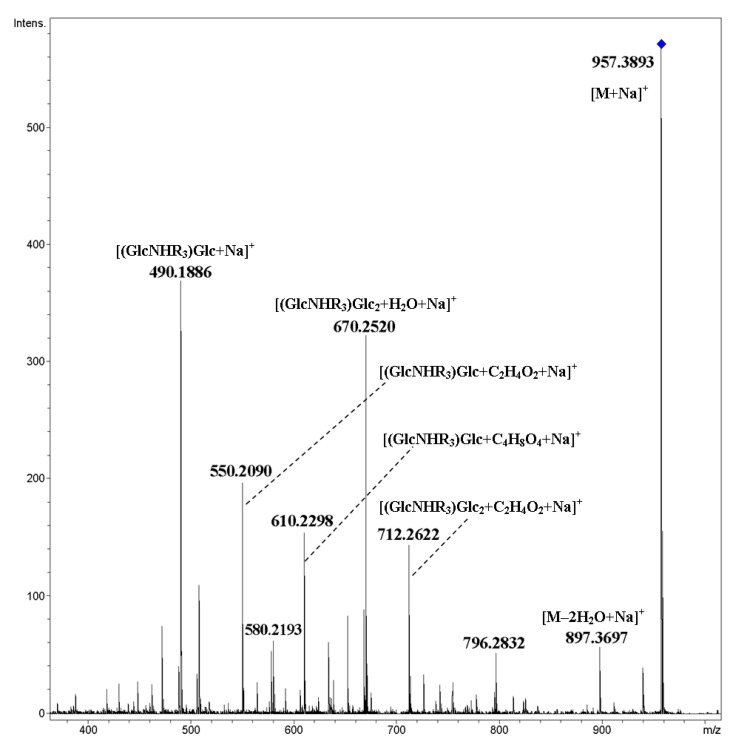
CID MS/MS of the [M+Na]^+^ ion of mixed, cyclic tetramer **13** (M 934.40 Da, R_3_ = CH_3_OC_4_H_8_OCH_2_CO), *m*/*z* 957, E_a_ 85 eV, tune wide acquisition method.

**Figure 12 ijms-21-08284-f012:**
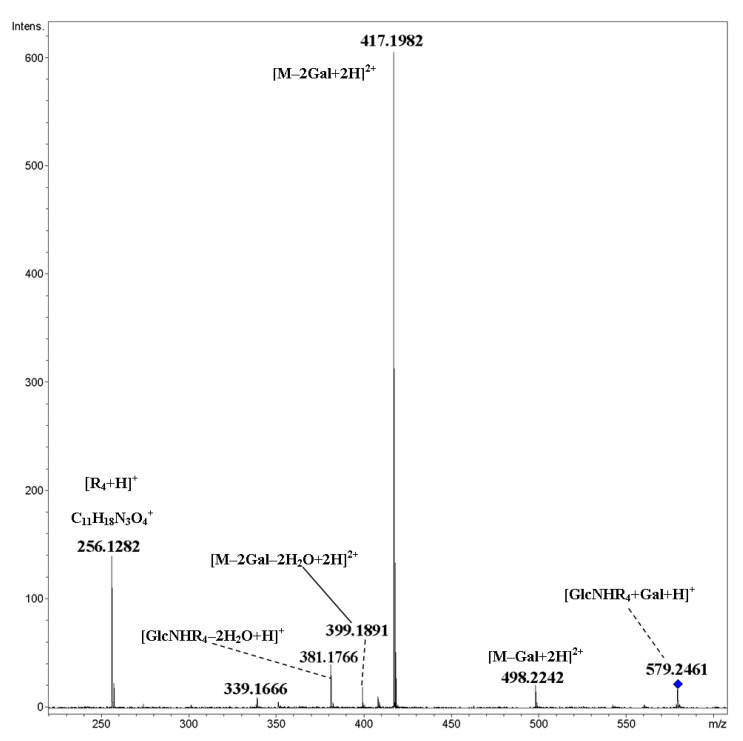
CID MS/MS of the [M+2H]^2+^ ion of cyclic dimer **14** (M 1156.49 Da, R_4_ = (OC_2_H_4_)_3_(C_2_HN_3_)CH_2_CH_2_(CO)-), *m*/*z* 579, E_a_ 25 eV, tune low acquisition method.

**Figure 13 ijms-21-08284-f013:**
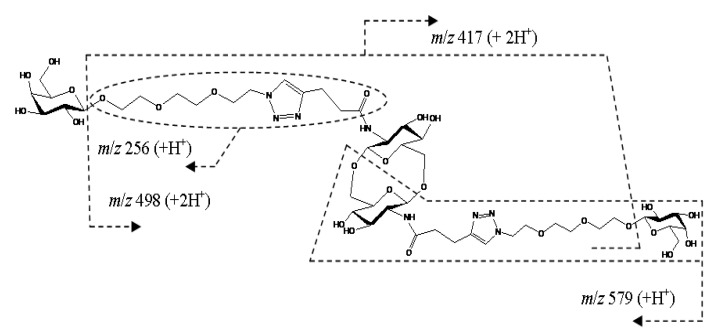
A possible fragmentation of the [M+2H]^2+^ ion of cyclic dimer **14** under CID. For MS/MS of **14**, see Figure 12.

**Figure 14 ijms-21-08284-f014:**
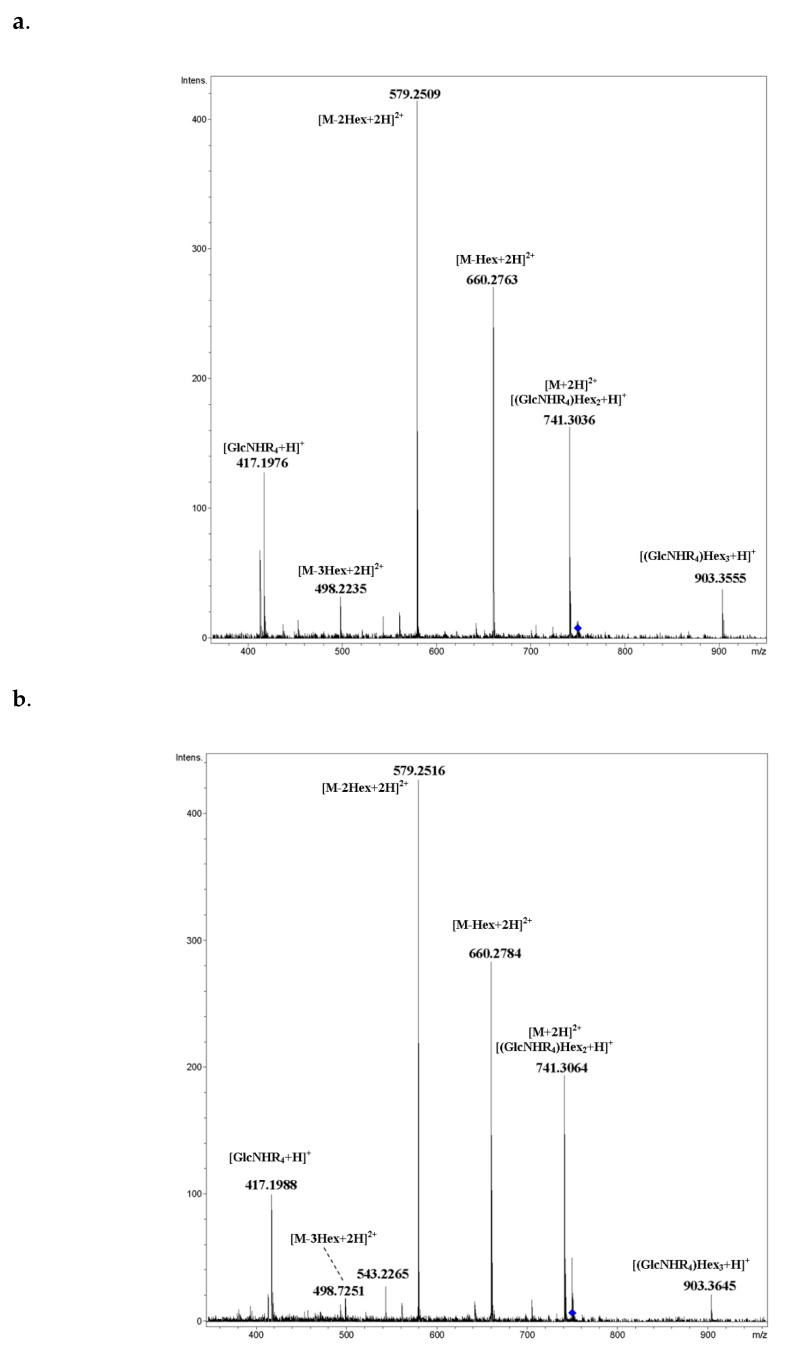
CID MS/MS of [M+H+NH_4_]^2+^ ions of isomeric mixed, galactosylated cyclic tetramers **15** (**a**) and **16** (**b**) (M 1480.59 Da, R_4_ = –(OC_2_H_4_)_3_(C_2_HN_3_)CH_2_CH_2_(CO)), *m*/*z* 750, E_a_ 25 eV, tune wide acquisition method. Hex: hexose residue, Glc or Gal.

**Figure 15 ijms-21-08284-f015:**
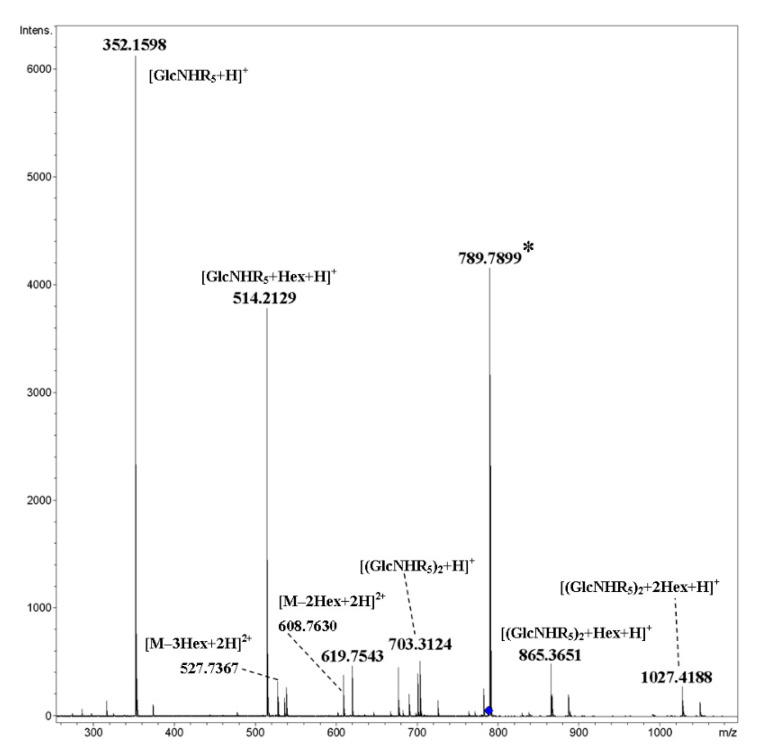
CID MS/MS of the [M+2NH_4_]^2+^ ion of cyclic, galactosylated trimer **17** (M 1539.62 Da, R_5_ = –(OCH_2_)_3_OCH_2_(CO)–), *m*/*z* 788, E_a_ 20 eV, tune low acquisition method. Hex: hexose residue, Glc or Gal. (*)—overlap of a residual peak of [M+H+K]^+^.

**Figure 16 ijms-21-08284-f016:**
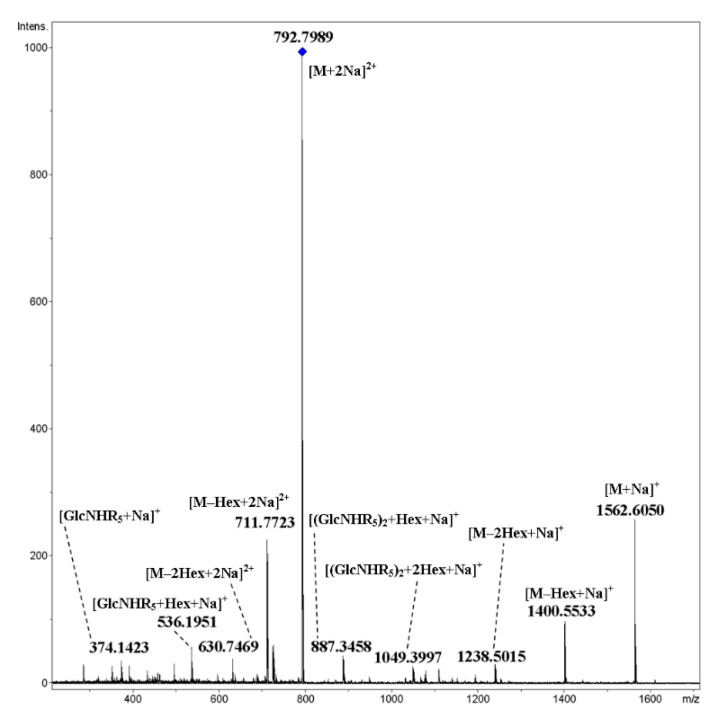
CID MS/MS of the [M+2Na]^2+^ ion of cyclic, galactosylated trimer **17** (M 1539.62 Da, R_5_ = –(OCH_2_)_3_OCH_2_(CO)–), *m*/*z* 792, E_a_ 50 eV, tune wide acquisition method. Hex: hexose residue, Glc or Gal.

**Figure 17 ijms-21-08284-f017:**
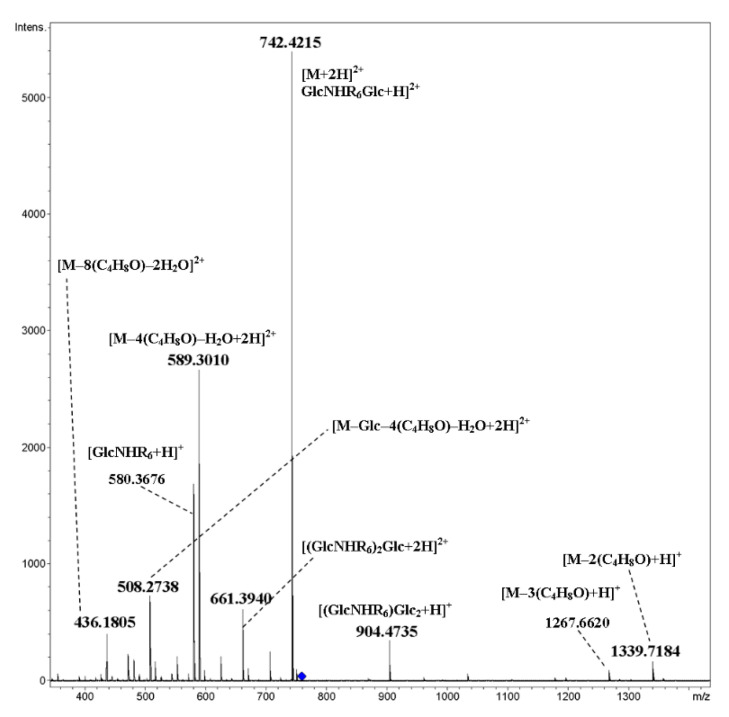
CID MS/MS of the [M+2NH_4_]^2+^ ion of mixed, cyclic tetramer **22** (M 1482.83 Da, R_6_ = H(OC_4_H_8_)_5_OCH_2_(CO)–), *m*/*z* 759, E_a_ 20 eV, tune wide acquisition method.

**Figure 18 ijms-21-08284-f018:**
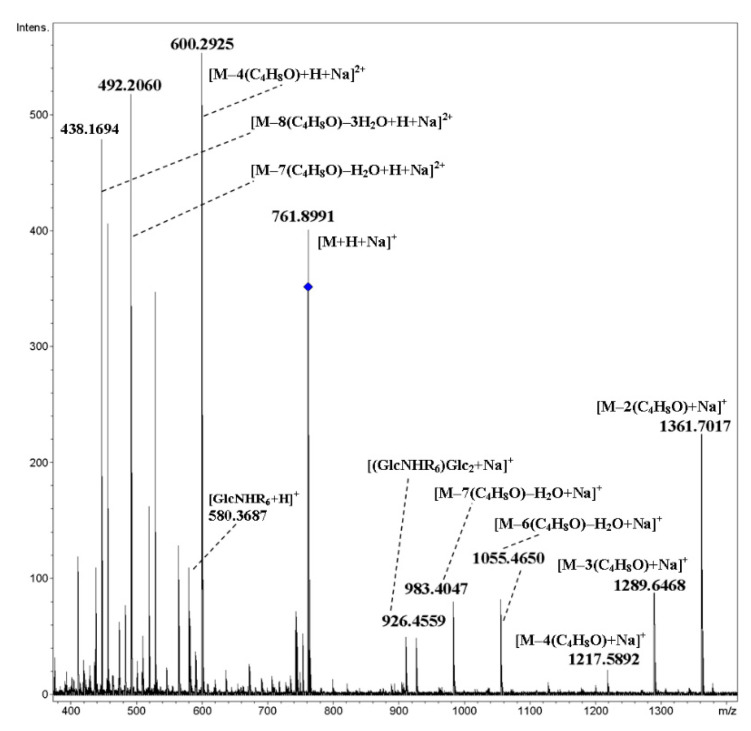
CID MS/MS of the [M+H+Na]^2+^ ion of mixed, cyclic tetramer **22** (M 1482.83 Da, R_6_ = H(OC_4_H_8_)_5_OCH_2_(CO)–), *m*/*z* 762, E_a_ 25 eV, tune wide acquisition method.

**Figure 19 ijms-21-08284-f019:**
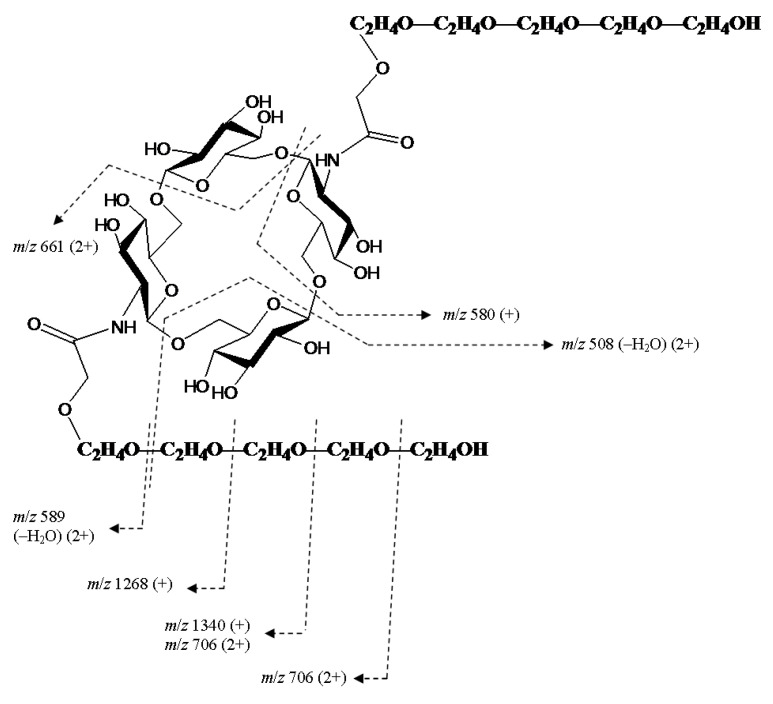
A proposed fragmentation of the [M+2NH_4_]^2+^ ion of mixed, cyclic tetramer **22** (M 1482.83 Da) under CID. For MS/MS of **22**, see Figure 17.

**Figure 20 ijms-21-08284-f020:**
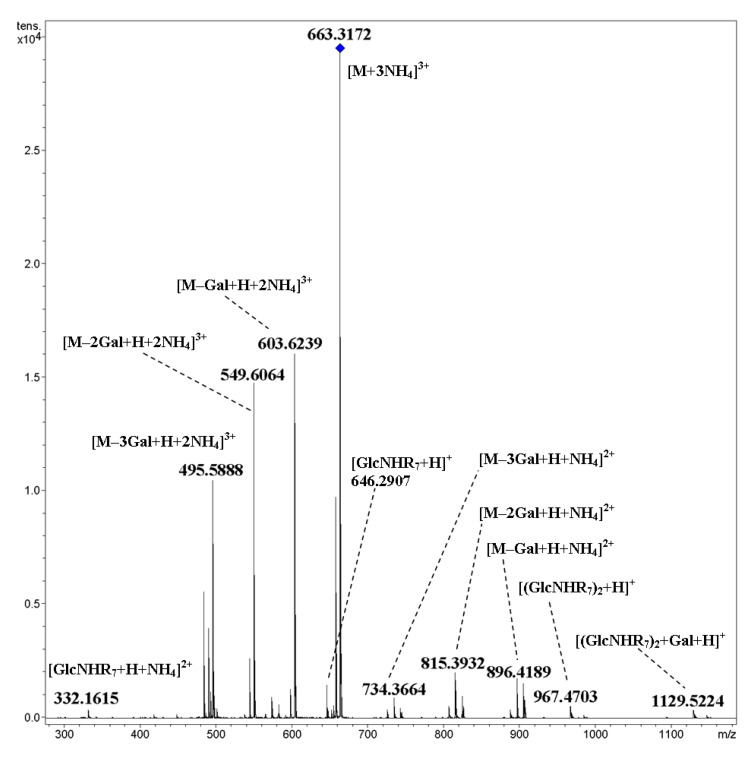
CID MS/MS of the [M+3NH_4_]^3+^ ion of cyclic, galactosylated trimer **23** (M 1935.85 Da, R_7_ = –(OC_4_H_8_)_5_OCH_2_(CO)–), *m*/*z* 663, E_a_ 12 eV, tune wide acquisition method.

**Figure 21 ijms-21-08284-f021:**
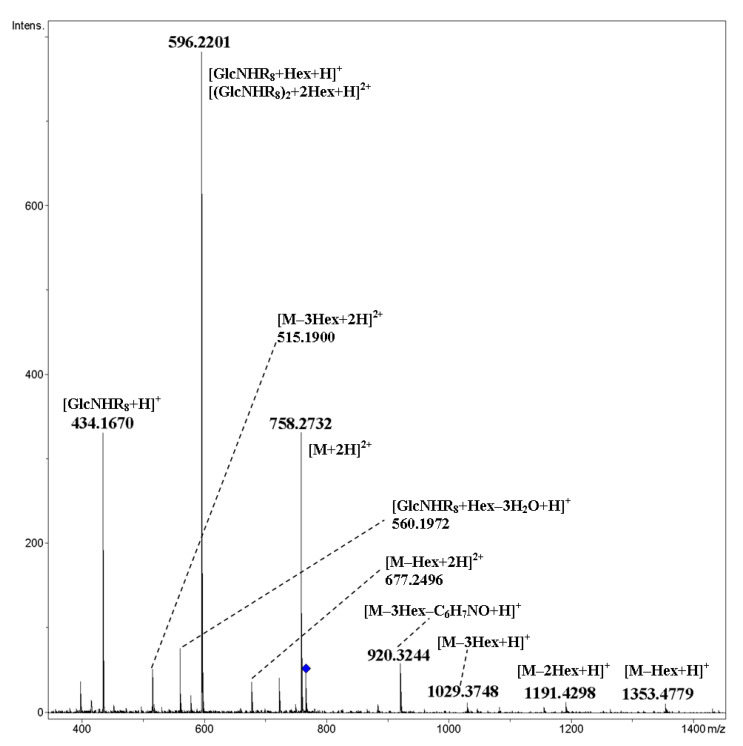
CID MS/MS of the [M+H+NH_4_]^2+^ ion of mixed, galactosylated cyclic tetramer **24** (M 1514.53 Da, R_8_ = –O(*p*-C_6_H_4_)NH(CO)CH_2_(C_2_N_3_)CH_2_(CO)–), *m*/*z* 767, E_a_ 25 eV, tune wide acquisition method. Hex: hexose residue, Glc or Gal.

**Figure 22 ijms-21-08284-f022:**
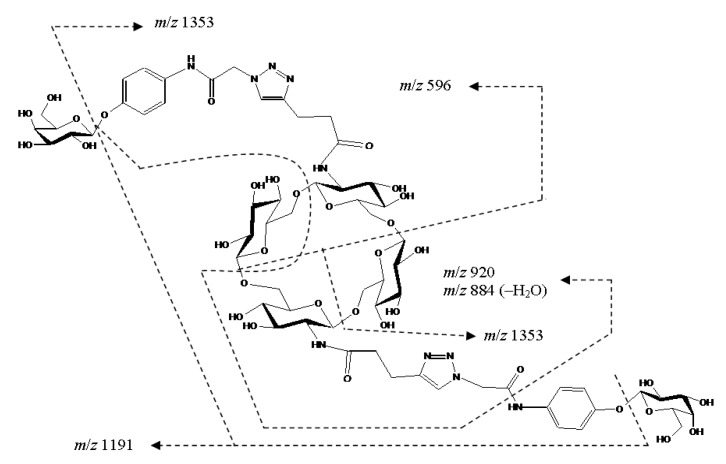
A proposed scheme of fragmentation of the [M+H+NH_4_]^2+^ ion of mixed, galactosylated cyclic tetramer **24** under CID. For MS/MS of **24**, see Figure 21.

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
