# Peer review of "Tandem Electrospray Mass Spectrometry of Cyclic N-Substituted Oligo-β-(1→6)-D-glucosamines"

_ijms, 2020, doi:10.3390/ijms21218284_

Round 1

Reviewer 1 Report

1. This is a good paper with high quality original research which is well designed and carried out.

2. The quality of English could be improved and there are

severalerrors of expression. For example,in line 20:“Cyclic oligosaccharides attract attention of researchers for a long time.” should be “Cyclic oligosaccharides has attracted theattention of researchers for a long time.” Line 32: “.....development of complementary approach based on mass spectrometry for investigation and/or support of the above structures seems actual.” should be written as “...........development of a complementary approach based on mass spectrometry for investigation and/or in support of the above structures is a realistic goal.”Line 48: “......such as cleavages of glycosidic bonds of cycles, residues in cycles and side chains, along with rearrangements was the aim of this study.” should be written as “.....such as cleavages of glycosidic bonds of the cyclic structure, residues in cyclic structuresand side chains, along with rearrangements was the aim of this study.”. There are other such errors which further proof reading should fix.

3.The proposed structures of the fragments make sense and are well presented.

4. Line 70 refers to Table S1 which cannot be found.

5. For the Figures, it would be easier for the reader to follow the fragments if the value of M is given.

6. For Figure 2, “Figure 2. CID MS/MS of the [M+2H] 2+i97 ion of cyclic tetramer 3” what is i?

7. In line 98 (Figure 2), “(R1= HCCCH2CH2CO)...’ should read “(R1107 = HCCCH2CH2CO)...’. Same comment applies to Figure 3 and other related figures.

Author Response

We are grateful to the Reviewer 1 for the constructive feedback of our manuscript "Tandem electrospray mass spectrometry of cyclic N-substituted oligo-beta-(1-6)-D-glucosamines submitted to the Special Issue "Complex Carbohydrates and Glycoconjugates: Structure, Functions and Applications" of the International Journal of Molecular Sciences. We address their comments point-by-point below, and we have highlighted changes within the manuscript using the track changes feature and yellow color.

Point 1. This is a good paper with high quality original research which is well designed and carried out.

Point 2. The quality of English could be improved and there are several errors of expression.

For example, in line 20: “Cyclic oligosaccharides attract attention of researchers for a long time.” should be “Cyclic oligosaccharides has attracted the attention of researchers for a long time.” 

Line 32: “…..development of complementary approach based on mass spectrometry for investigation and/or support of the above structures seems actual.” should be written as “………..development of a complementary approach based on mass spectrometry for investigation and/or in support of the above structures is a realistic goal.” 

Line 48: “……such as cleavages of glycosidic bonds of cycles, residues in cycles  and side chains, along with rearrangements was the aim of this study.” should be written as “…..such as cleavages of glycosidic bonds of the cyclic structure, residues in cyclic structures and side chains, along with rearrangements was the aim of this study.”. 

There are other such errors which further proof reading should fix. 

Responses.

Line 20. Accepted.

Line 32 (36). Accepted.

Line 48 (51). Accepted.

Point 3. The proposed structures of the fragments make sense and are well presented.

Point 4. Line 70 refers to Table S1 which cannot be found.

Response.

Line 73. Added: "Supplementary Materials, " etc.

Point 5. For the Figures, it would be easier for the reader to follow the fragments if the value of M is given.

Response.

For all of the mass spectral figures, molecular masses of neutrals (M) were added for the studied compounds in figure legends.

Point 6. For Figure 2, “Figure 2. CID MS/MS of the [M+2H]2+i 97 ion of cyclic tetramer 3” what is i?

Response: "i" is a misprint. Deleted.

Point 7. In line 98 (Figure 2), “(R1 = HCCCH2CH2CO)…’ should read “(R1 107 = HCCCH2CH2CO)…’. Same comment applies to Figure 3 and other related figures.

Response: In all of the relevant figure legends, superscripts were changed to subscripts. Accordingly, changes were done in the text (lines 91—92, etc.).

Reviewer 2 Report

The English needs to be addressed. Please double check the correct/common English terminology for scientific terms. Additionally, I found the writing in the paper hard to follow. It would be helpful to clearly reference which figure is being discussed in the results and discussion. It would be helpful to cross reference figures of fragmentation schemes with the corresponding spectra.

Some of the figures (13, 22) have poor resolution and should be re-drawn.

It would be helpful to define what some of the terms specific to the instrument mean like "tune wide acquisition method" so that someone with a different instrument can understand the instrumental conditions.

Author Response

We are grateful to the Reviewer 2 for the constructive feedback of our manuscript "Tandem electrospray mass spectrometry of cyclic N-substituted oligo-beta-(1-6)-D-glucosamines submitted to the Special Issue "Complex Carbohydrates and Glycoconjugates: Structure, Functions and Applications" of the International Journal of Molecular Sciences. We address their comments point-by-point below, and we have highlighted changes within the manuscript using the track changes feature and bright blue color.

Point 1. The English needs to be addressed. Please double check the correct/common English terminology for scientific terms.

Response:

We have checked common English and scientific terminology and corrected writing of the text and figure legends (lines 22, 39, 45, 46, 163, 169, 181, 182, 190, 196, 250, 288, 303, 319).

Point 2. Additionally, I found the writing in the paper hard to follow. It would be helpful to clearly reference which figure is being discussed in the results and discussion.

Response:

We have added to the manuscript missed references to Figs. 8 and 12, respectively (lines 159 and 193). Two references were added to describe the problem of the study more thoroughly (lines 24—25).

Point 3. It would be helpful to cross reference figures of fragmentation schemes with the corresponding spectra.

Response:

We have added cross references in Figs. 13, 19, and 22 legends (lines 203—204, 246 and 273, resp.).

Point 4. Some of the figures (13, 22) have poor resolution and should be re-drawn.
Response:

We have redrawn Figs. 13 and 22 using tiff format.

Point 5. It would be helpful to define what some of the terms specific to the instrument mean like "tune wide acquisition method" so that someone with a different instrument can understand the instrumental conditions.

Response:

In Materials and Methods (lines 305—307):

Standard tunes of orthogonal accelerator and other parts were set with a tune wide method for middle and high m/z values (higher than 300 Th), and a tune low method for low ones, m/z from 50 to 300 Th.

Tune parameters are tightly bound with a type and even a model of the instrument used. We give some of them in Materials and Methods. Really, exact values of all acquisition parameters for “tune wide”, “tune low” and other so-called acquisition "methods" are hidden for ordinary users (operators). At least, there is no information about the choice of all voltages on electrodes of ion optics, ion transfer lines, multipoles, an orthogonal accelerator and a reflectron in the user's manual. It is quite possible that this info (at least in part) is know-how of the manufacturer (like software as whole). Nevertheless, the MS patterns strongly depend on the method used (despite of it is commonly ignored), so, we assume that it is necessary to give "methods" used for better reproducibility.